# Self-Perception of the Digital Competence of Educators during the COVID-19 Pandemic: A Cross-Analysis of Different Educational Stages

**Javier Portillo *** [ID]**, Urtza Garay** [ID]**, Eneko Tejada** [ID] **and Naiara Bilbao** [ID]

Department of Didactics and School Organization, Faculty of Education, University of the Basque Country (UPV/EHU), 48940 Leioa, Spain; urtza.garay@ehu.eus (U.G.); eneko.tejada@ehu.eus (E.T.); naiara.bilbao@ehu.eus (N.B.)

**\*** Correspondence: javier.portillo@ehu.eus; Tel.: +34-946-017-534

**Abstract:** The objective of this research is to measure the perception that teachers had about their own performance when they were forced to carry out Emergency Remote Teaching due to the COVID-19 pandemic. A questionnaire was provided to teachers of every educational stage in the Basque Country (Pre-school, Primary and Secondary Education, Professional Training, and Higher Education) obtaining a total of 4586 responses. The statistical analysis of the data shows that the greatest difficulties reported by educators are shortcomings in their training in digital skills, which has made them perceive a higher workload during the lockdown along with negative emotions. Another finding is the existing digital divide between teachers based on their gender, age, and type of school. A further worrying result is the lower technological competence at lower educational levels, which are the most vulnerable in remote teaching. These results invite us to reflect on the measures to be taken to improve equity, social justice, and the resilience of the educational system, which align with some of the Sustainable Development Goals.

**Keywords:** digital technology; inclusive education; b-learning; educational policy; educational organization

## 1. Introduction

The Sustainable Development Goals (SDGs) are a universal call to action to end poverty, protect the planet, and improve the lives and prospects of everyone, everywhere. The 17 Goals were adopted by all UN Member States in 2015 as part of the 2030 Agenda for Sustainable Development, whic set out a 15-year plan to achieve the Goals. Although remarkable progress has been made, overall, action to meet the Goals is not yet advancing at the speed or scale required. The UN claimed a decade of action: "2020 needs to usher in a decade of ambitious action to deliver the Goals by 2030" [1]. Instead, the unexpected emergence of COVID-19, and the resulting global confinement since March 2020, has contributed to hindering its development [2].

COVID-19 has broadened existing inequalities because, for the most vulnerable communities, it has severely affected their health, economy, and education. The pandemic has made more visible how communities with low economic resources and fragile social protection nets suffer to a greater extend the consequences of the crisis. Social, political, educational, and economic inequalities have amplified the effects of the pandemic [2]. Among the most affected vulnerable communities are low-income families and women. Therefore, COVID-19 has directly influenced the delay in the progress made on Goal 5, "Gender Equality", and Goal 4, "Quality Education". At the same time, Quality Education during confinement has been intimately linked to the need for development of Goal 9, "Building resilient infrastructure, promoting sustainable industrialization, and encouraging innovation".

Information and Communication Technologies have been at the forefront of the response to COVID-19. The crisis has accelerated the digitization of education, but it has also contributed to increasing the digital divide among students [3–5], which has been dragging on for years [6,7].

Thus, according to the United Nations, in 2020, 3.6 billion people still do not have an internet connection and cannot access online education.

Emergency Remote Teaching [8,9] was developed as a rapid response [10–13] to the situation. Its nature has made the proper acquisition and access to the needed technology difficult [14]. The reduction of present and, above all, future inequalities demands that every student and teacher is suitably trained to acquire the digital competences they need in digital environments [15]. We must remember that Emergency Remote Teaching (ERT) is an alternative way of teaching due to the circumstances of crisis [16–21], while quality online or blended learning ask for careful instructional design and planning. Online learning has proven its effectiveness in numerous research studies [22–24] whenever a systematic model for design and development is adopted [25]. Therefore, in order to contribute to the improvement of the quality of education, it seems necessary to carry out an in-depth analysis of what has been done and what should be improved. This unexpected challenge has placed on the agendas of the major educational leaders the need to develop the digitalization of education (SDG 9) and hopefully reduce the digital divide, the social gap, and gender inequality in the population (SDG 10) as one of the axes to guarantee a quality education (SDG 4).

On this shift of education to digital, the European Union has published the Digital Education Action Plan (2021–2027), which sets out the criteria for high-quality, inclusive, and accessible digital education for all in Europe. The plan aims to make education and training systems fit for the digital age and presents two strategic priorities: fostering the development of a high-performing digital education ecosystem and enhancing digital skills and competences for the digital transformation [26]. In short, it requires work to be carried out on infrastructure, connectivity, and digital equipment, and also on the development of digital literacy, which will mean breaking down the inequalities among the population.

The "Digital Spain Plan 2025" [27] is aligned with the European Commission in promoting the digitization of education with a radical change in methods and contents, including the promotion of distance education and the implementation of digital vouchers to facilitate connectivity for students. Therefore, in addition to supplying technological resources to the classrooms, the development of the digital competence of students and teachers is prioritized in order to reduce the digital divide.

Digital Competence of Educators (DCE) is not a new concept, and it has been studied by researchers of educational technology over the last few decades [28–35]. In the European arena, the European Framework for the Digital Competence of Teachers (DigCompEdu) [36] is a scientifically sound framework describing what it means for educators at all stages to be digitally competent. DigCompEdu details 22 competences organized in six Areas and distinguishes six levels along which educators' digital competence typically develops. For each of the 22 competences, level descriptors and proficiency statements are provided and allow educators to understand their level of competence and their specific development needs. The framework aims to detail how digital technologies can be used to enhance and innovate education and training.

The impact of the pandemic and, above all, the period of severe confinement experienced since March 2020 has forced the abrupt development of the DCE for active teachers. But it has also brought to the table the gaps that exist between teachers and the needs to develop some key aspects of DigCompEdu. Prior to this situation, teachers had developed their DCE in their daily interaction with technology [37], but this new situation has pushed them to increase the use of digital resources abruptly in order to respond to the change in the reality in which the teaching-learning process is taking place. At pointed out by [38], teaching professionals have gained in fluency, mastery, and comfort, not only in the use of basic applications but also in the management of information, the creation of content, and the use of technology to keep their students connected.

This paper presents a study based on the analysis of the self-perception teachers have of their digital competence; that is, how their proficiency level of digital competence has influenced the

development of quality Emergency Remote Teaching during the COVID-19 confinement. Therefore, it contributes to the measurement of the DCE at a time when DCE constrained teachers' daily work at all educational levels and the resilient response of the educational system to a situation never experienced before.

## 2. Materials and Methods

The main objective of this research is to analyze whether the teaching community has perceived itself as digitally capable of dealing with Emergency Remote Teaching (ERT) caused by the COVID-19 pandemic. Having this objective in mind, the following research questions have been posed:

(1) Have teachers perceived themselves as digitally competent to deliver Emergency Remote Teaching (ERT)?
(2) Is Digital Competence of Educators (DCE) biased depending on gender, age, educational institution, or stage of education?
(3) Did the training received in DCE have an impact in the perception of well-being of teachers during Emergency Remote Teaching?

To this end, the following variables have been considered: Digital Competence of Educators (DCE), DCE Training, workload of teachers (before and during the lockdown), and emotions (positive and negative) experienced during the confinement. The impact of these other secondary variables have also been taken into account: age, gender, type of educational institution (private or public), and educational stage.

### 2.1. Design of the Questionnaire

Four blocks of questions were defined in order to measure the Digital Competence of Educators (DCE), their workload before and during the lockdown, the (positive and negative) emotions they experienced during confinement, and socio-demographical data.

Regarding DCE, some of the existing questionnaires about the digital skills of teachers [39,40] were adapted to the context of Emergency Remote Teaching. The result was a questionnaire composed of seven multidimensional Likert-type scales, with response options from 1 (little) to 5 (a lot); two multiple-choice questions; two questions with a dichotomous answer (Yes/No); and one open question. The first five Likert scales were used to measure the perception of their digital skills. The DigCompEdu framework and the National Institute of Technology and Professional Development (INTEF) [41] proposal on the development of Digital Competences for teachers were taken into account. Two other scales registered the perception of the ability of students to deal with emergency remote learning and the quality of the training that teachers had received in digital skills. Finally, a question about whether some kind of 'emergency' training had been delivered during the confinement period was included.

In relation to the workload, the questionnaire developed by [42] and validated by [43] was adapted to the context of the Autonomous Community of the Basque Country (ACBC) and the pandemic situation. The questionnaire consisted of six multidimensional Likert-type scales, with response options from 1 (low) to 5 (high). The first three scales referred to the demands imposed on the person (mental, physical, and temporal demand) and the other three referred to the person's interaction with the task (effort, performance, and frustration).

The block about emotional responses consisted of items to evaluate positive emotions (pride, satisfaction, enthusiasm, confidence, and relief) and items to measure negative emotions (insecurity, stress, concerns, anger, and frustration) [44]. The instrument used was generated by adapting the questionnaire for virtual learning environments (WebCT) proposed by [45].

The questionnaire, the dataset, the 4589 responses, and supplementary research data have all been publicly shared online [46].

## 2.2. Population

A non-probabilistic sampling method was selected attending to the limitations to access the target population [47]. Specifically, a convenience sampling was carried out by sending the questionnaire by email to all educational centers of the Autonomous Community of the Basque Country (ACBC) in two versions using Basque and Spanish languages. The 'snowball' technique was also used for non-probabilistic sampling in conditions of difficult access [48], making use of social networks (Facebook and Twitter) and other teaching networks articulated through WhatsApp groups. Information was collected during three weeks in May 2020. Finally, 4589 responses to the questionnaire were obtained; that is, almost 10.5% of all the teaching staff of the Basque Country [49] participated voluntarily in the study. Regarding internet connectivity, 1.5 million persons (80.2% of the population) in the Basque Country have a broadband connection [50].

An ad hoc questionnaire with four questions (age, gender, educational stage of teaching, and type of educational center) was used to find out the socio-demographic profile. Out of the 4589 teachers in the sample for this study, 23.3% are male, 75.5% female, and 0.2% non-binary, with an average age of 54 (SD = 6.24). These professionals work at different educational stages (Pre-school Education, 10.8%; Primary Education, 31.6%; Secondary Education and Baccalaureate, 38.3%; Vocational Training, 5.3%; Higher Education, 8.6%; Others, 5.4%) and in different types of educational centers (public centers, 77.2%; private centers, 22.8%).

## 2.3. Validation of the Questionnaire

The psychometric properties of the questionnaire were analyzed. Firstly, the total sample was randomly divided into two halves. With the first subsample (*n* = 2296), a Parallel Analysis (PA) was carried out in order to explore the factorial structure of the instrument. In this case, the software Factor 10.4.01 [51] was used. The procedure selected to determine the number of dimensions was the optimal implementation of the PA [52], and the parameter estimation method used was the Diagonally Weighted Least Squares (DWLS) method. This method is the most appropriate when the variables are ordinal, as in the case of Likert-type scales [53]. Finally, taking into account that a one-dimensional solution was expected, no rotation method was applied. Thus, the results of this first analysis suggested a unifactorial structure for five items of the DCE block, two factors for perceived workload before and during lockdown, and two factor for negative and positive perceived emotions.

Based on these first exploratory results, a Confirmatory Factorial Analysis was carried out with the second subsample (*n* = 2294). This analysis was performed with Lisrel 8.80 software [54], using the DWLS parameter estimation method. The quality of the fit was evaluated through the following goodness-of-fit indexes: the Root Mean Square Error of Approximation (RMSEA), whose value must be less than 0.08 [55], the Non-Normed Fit Index (NNFI), the Comparative Fit Index (CFI), and the Goodness of Fit Index (GFI), whose values must be greater than 0.95 [56]. The results of this analysis showed a satisfactory fit of the two-factor model to the data: RMSEA = 0.04, NNFI = 0.99, CFI = 0.99, GFI = 0.99

## 2.4. Statistical Analysis

The statistical values of the responses to each of the five items included in the Digital Competence of Educators (DCE) factor were calculated, as were those regarding the question about the educational platform used. Next, the changes in the DCE factor were analyzed by means of ANOVA according to the socio-demographic variables: gender, age, type of educational center (public, private), and educational stage. The relationships between DCE proficiency and three other items—training in DCE, performance of students online, and use of educational platforms—were also analyzed. This was implemented through Spearman's non-parametric correlation coefficient. Pearson's method was applied to identify the correlations between perceived workload factors (during and previous to the confinement), perceived emotions (positive and negative), and digital competence.

Finally, by means of two hierarchical regressions, to what extent the workload before and after the lockdown and the DCE variable predict the appearance of positive emotions (regression 1) and negative emotions (regression 2) was analyzed. In both cases, the socio-demographic variables and the opposite emotion variable—negative emotions in the case of regression 1 and positive emotions in regression 2—were controlled as covariates. It was implemented in a two-step process, where the first step included the covariant variables (whose effect we are interested in controlling) and the second step added the predictor variables (workload before and during the confinement and DCE). This procedure allows us to determine to what extent an association of variables predicts positive or negative emotions, extending the one-to-one variable analysis performed by correlation.

## 3. Results

This section describes the results of the analysis, attending to each of the three research questions.

### 3.1. Have Teachers Perceived Themselves as Digitally Competent to Deliver Emergency Remote Teaching (ERT)?

In relation to the first research question, teachers have perceived themselves to be partially competent of developing ERT. Moreover, they think they are more skilled in the use of digital tools for general communication but they feel less confident concerning the specific tools used to facilitate teaching-learning processes.

The distributions of responses to statements about DCE are very symmetrical, fitting well into the normal distribution. The statement "I have the knowledge and skills to use online communication tools (chat, forum, videoconference, e-mail...)" stands out from the rest because it has the highest mean (4.00 out of 5), the lowest standard deviation (less dispersion), the highest negative asymmetry (distribution tail lengthens for values below the mean), and the only positive kurtosis (data concentrated on the mean, pointed curve). "I have the knowledge and skills to use the educational platform" scores 0.53 points less, which is a significant difference (13.25%), while "I have basic knowledge and skills to create and edit online activities" has a mean value of 3.60; "I have basic knowledge and skills to search for online activities" has a value of 3.70; and "During the confinement I have had difficulty in making corrections and getting them to the students" has 3.00.

The averages in the DCE factor, which comprises the previous five statements, were analyzed according to the educational platform each teacher used. The highest value, 1.13 points above the average and with the smallest deviation, is given for Moodle, followed by Google Classroom, which is the most widely used platform. The lowest value is given among Microsoft users. It is significant that 14.9% do not use any platform despite the fact that the use of an educational platform has shown a strong correlation (0.40) with the performance, perceived by their teachers, of students in online learning and with the DCE factor (0.29).

In addition, DCE is also related to the existence or absence of training during confinement as well as to the familiarity with digital educational platforms before the pandemic. This has been studied by means of the correlation of the DCE factor with the answers to the questions "Have you received training in digital skills and educational integration of technologies?" resulting in a correlation coefficient of 0.31 and "In the COVID19 period, have you received training to adapt your subjects to digital format?" scoring 0.14. This seems to point out that emergency training in digital literacy is good for DCE, but it is far more effective the training received before the emergency. In the same vein, there is a correlation (0.27) between the use of educational platforms and the quality of the previous training received.

What is also very significant (0.36) is the correlation between the DCE factor and online performance of students.

### 3.2. Is DCE Biased Depending on Gender, Age, Educational Institutio, or Stage of Education?

The second research question inquires about the existence of gaps in DCE between teachers according to their age, gender, type of school (public, private), or educational stage.

With regard to gender, the results showed a significant effect, $F_{(2, 4588)} = 24.97$, $p < 0.001$, with men (M = 12.46) scoring higher than women (M = 11.55) in DCE factor. The data show a gender digital gap quantified at 0.91 points, with the number of women in the sample more than triple that of men.

In relation to age, a significant effect was also found, $F_{(7, 4588)} = 32.46$, $p < 0.001$. Hochberg's Post Hoc GT2 test generally showed that older teachers were less technologically competent than younger ones. There is a marked linear decrease in the DCE mean value according to age, with the difference being quantified at 3.26 points (from 13.66 in the 21–25 years range to 10.40 in 61–65). The standard deviation also increases with age (3.22 points, from 3.030 to 4.250), showing that the DCE is more homogeneous among the youngest teachers. It should also be noted that the most numerous group (41–50) is 28.7% of the total, and its DCE corresponds to the total average.

With regard to the type of school, a significant effect was also found, $F_{(2, 4585)} = 5.54$, $p = 0.004$. Hochberg's Post Hoc test showed that public schools (M = 11.66) scored lower in technological competence than private schools (M = 12.12). The digital divide by type of center is quantified in a difference of 0.46 points in DCE mean between public and private centers. Bearing in mind that half of all students in the Autonomous Community of the Basque Country (ACBC) study in mixed schools [57], participation in the sample has been overwhelmingly greater among public teachers than among teachers from private centers despite the fact that the questionnaire was sent to all school management bodies in the ACBC.

Regarding the educational stage, a significant effect of $F_{(6, 4559)} = 48.82$, $p < 0.001$ was observed. Hochberg's Post Hoc GT2 test revealed that, generally speaking, the higher the grade, the higher the digital competence.

There is a linear increase in the DCE mean value as the educational grade grows, with a difference of 2.65 points (from 10.46 in Early Childhood education to 13.11 in University education), with the standard deviation also being lower as the educational level increases.

### 3.3. Did the Training Received in DCE Have an Impact in the Perception of Well-Being of Teachers during Emergency Remote Teaching?

The results also show that there is a relationship between DCE and the perception of well-being in the development of ERT.

As well as the DCE factor (questions about digital competence), which was composed of five items, other factors were created by the association of items related to the workload of teachers before and during the confinement and the emotions, positive and negative, experienced during the lockdown. Table 1 shows the result of analyzing all the correlations between these five factors using Pearson's correlation coefficient.

**Table 1.** Correlation coefficient between variables (Pearson's r).

|  | 1 | 2 | 3 | 4 | 5 |
|---|---|---|---|---|---|
| 1. Pre-COVID workload factor | - |  |  |  |  |
| 2. COVID workload factor | 0.18 ** | - |  |  |  |
| 3. Positive Emotions factor | 0.17 ** | 0.12 ** | - |  |  |
| 4. Negative Emotions factor | 0.02 | 0.47 ** | −0.32 ** | - |  |
| 5. Digital Competence of Educators (DCE) factor | 0.05 ** | −0.03 * | 0.31 ** | −0.25 ** | - |

\* $p < 0.05$, \*\* $p < 0.001$.

Among the results, it is noticeable that, due to its high level of relevance, negative emotions are strongly related to high workloads during COVID (0.47) and better training in CDE is associated with more positive emotions (0.31) and less negative emotions (0.25). Furthermore, in some cases,

a good training in digital competences is slightly correlated with less workload during COVID (−0.03). Therefore, teachers who perceive themselves as digitally competent present more positive emotions, and some of them feel that they have had a smaller workload.

On the basis of meaningful relations among variables, a hierarchical regression can be designed in order to confirm the findings [58–60]. Table 2 revels the results of a two-step hierarchical regression, which predicts positive emotions from workload variables and DCE—dependent variables—controlling, at the same time, as covariates, the effect of socio-demographic variables and negative emotions.

**Table 2.** Hierarchical regression analysis of predictors of positive emotions.

| Measurement | B | E.T.B | β |
|---|---|---|---|
| **Step 1—Covariate Variables** | | | |
| Educational stage | 0.03 | 0.04 | 0.01 |
| Type of center | 0.40 | 0.11 | 0.05 *** |
| Gender (1 = man; 2 = woman; 3 = non binary) | 0.07 | 0.13 | 0.01 |
| Age | 0.11 | 0.03 | 0.05 ** |
| Negative emotions | −0.28 | 0.01 | −0.32 *** |
| **Step 2—Workload + competencies** | | | |
| Educational stage | −0.05 | 0.03 | −0.02 |
| Type of center | 0.22 | 0.10 | 0.03 * |
| Gender (1 = man; 2 = woman; 3 = non binary) | −0.23 | 0.12 | −0.03 * |
| Age | 0.19 | 0.03 | 0.08 *** |
| Negative emotions | −0.35 | 0.01 | −0.40 *** |
| Workload before lockdown | 0.15 | 0.02 | 0.11 *** |
| Workload during lockdown | 0.39 | 0.02 | 0.29 *** |
| Competencies | 0.24 | 0.01 | 0.23 *** |

* $p < 0.05$; ** $p < 0.01$; *** $p < 0.001$. Positive emotions: $R^2 = 0.11$ ($p < 0.001$) in step 1; $\Delta R^2 = 0.15$ ($p < 0.001$) in step 2. $R^2 = 0.26$.

Hierarchical regression should be understood as a framework for model comparison [61]. In this sense, the prediction model that considers workloads and competencies as predictors is meaningfully better ($\Delta R^2 = 0.15$) than the first one, which is made of covariates. All predictor variables and most covariates have a significant effect on the prediction of positive emotions. Negative emotions are the most influential, also with a significant weight on the COVID workload and the DCE. Covariates have little influence, so they are not relevant for predicting positive emotions. In conclusion, high DCE and absence of negative emotions could predict positive emotions, and it is significant that high workload can also be present in the mix.

On the other hand, Table 3 presents the results of the two-step hierarchical regression that predicts negative emotions from the workload and DCE variables, controlling, in turn, as covariates, the effect of socio-demographic variables and positive emotions.

The model explains 38% (total $R^2$) of the variance shared by all variables together, which is higher than in the case of positive emotions (26%). The influence of competencies and workload predictors ($\Delta R^2 = 0.25$) is nearly double that of the covariates ($R^2 = 0.13$). The most relevant variable in the prediction is by far the COVID workload, with the DCE also having a significant weight and a significant effect of the covariate 'positive emotions'.

Therefore, the hierarchical regression confirms that negative emotions can be predicted by high workload and low DCE, as well as the absence of positive emotions. This combination is even worse in the case of women.

**Table 3.** Hierarchical regression analysis of predictors of negative emotions.

| Measurement | B | E.T.B | β |
|---|---|---|---|
| **Step 1—Covariate variables** | | | |
| Educational stage | −0.17 | 0.04 | −0.06 *** |
| Type of center | 0.07 | 0.13 | 0.01 |
| Gender (1 = man; 2 = woman; 3 = non binary) | 1.27 | 0.14 | 0.13 *** |
| Age | 0.13 | 0.04 | 0.05 ** |
| Positive emotions | −0.37 | 0.02 | −0.32 *** |
| **Step 2—Workload + Competencies** | | | |
| Educational stage | −0.09 | 0.03 | −0.03 ** |
| Type of center | −0.02 | 0.11 | −0.01 |
| Gender (1 = man; 2 = woman; 3 = non binary) | 0.18 | 0.12 | 0.02 |
| Age | −0.03 | 0.03 | −0.01 |
| Positive emotions | −0.39 | 0.02 | −0.34 *** |
| Workload before lockdown | −0.02 | 0.02 | −0.01 |
| Workload during lockdown | 0.78 | 0.02 | 0.50 *** |
| Competencies | −0.14 | 0.02 | −0.12 *** |

** $p < 0.01$; *** $p < 0.001$. Negative Emotions: $R^2 = 0.13$ ($p < 0.001$) in step 1; $\Delta R^2 = 0.25$ ($p < 0.001$) in step 2. $R^2$: 0.38.

## 4. Discussion

The pandemic has exposed the fact that educational systems around the world should improve their resilience to unexpected situations for the effective development of "Quality Education" (SDG 4), even in the event of those situations. Regarding educators and the goals of "Good health and well-being" (SDG 3) and "Decent work" (SDG 8), this research shows they have experienced high workload, stress, and negative emotions during the ERT. This study also verifies the fact that digital gaps have flourished in this emergency situation. Education systems should guarantee that the digital divide is reduced at all stages of education, which is linked to the goals of "Gender equality" (SDG 5) and "Reduced inequalities" (SDG 10).

The need for proper Digital Competence for Educators has been fundamental for the avoidance of disruption in teaching-learning processes. The results seem to draw a pattern: weaknesses in DCE increase when it comes to situations or tools specifically related to online teaching [62]. In other words, competence is greater in the regular digital communication skills (chat, forum, videoconference, email...) that most people usually use, regardless of their profession. This is an important nuance because these are the specific digital skills needed for the development of teaching methods (creating and managing meaningful activities online, knowing how to use the educational platform, structuring a subject online, etc.) that prove to be more related to good student performance [63,64]. This leads to the need for suitable training on DCE in a structured manner. Several studies [65,66] have addressed this demand for future teachers at faculties of education, as well as the demand for active teachers by means of Professional Development programs. It can be said that COVID-19 has emphasized the importance of teacher professional development for online and blended learning [67].

In the same vein, there are three other digital gaps that the results reveal: the gender gap between active teachers, the age gap, and the gap that arises in relation to the type of educational center (private and public).

In terms of the gender gap, this study has pointed out the lower mean values in DCE for female teachers, as well as the greater likelihood of suffering the mix of negative emotions and high workload during the confinement. The data from this study corroborate what has been observed in other geographical areas [68,69], as well as in other spheres outside of education [70]. In line with this, the European Commission's publication "Women in the Digital Scoreboard" shows that Spain is in a low position in all the indicators with regard to all types of skills associated with Information and Communication Technologies (ICT) and that the difference between genders is very significant, clearly establishing a difference in favor for men in all the skills analyzed [71]. These data call for

urgent action to put women on an equal footing in the so-called fourth industrial revolution. It is perhaps an indicator that, even within the education system, the transmission of social gender roles is maintained and that there is still a lack of role models to contribute to the equal use of ICTs.

An age gap regarding digital issues has also been detected among active teachers in the Autonomous Community of the Basque Country during the pandemic. The need for continuous training has been confirmed because, as [72] point out, the progressive increase in the age divide has a great impact on the instruction of students for full personal and professional life in a world where technology has taken on special relevance. Equal opportunities come hand in hand with digital literacy of our students [73]. This article presents results based on the self-perception of teachers and this fact leaves room for interpretation. Maybe older teachers feel that their tech expertise is not sufficient for them to develop in a digital environment the complexity of inquiry and project instruction that they can manage in a face-to-face environment thanks to their experience. This opens an interesting line for future research work.

Finally, with regard to the differences detected according to the type of the educational center, this result can be compared with other studies at an international level that show similar conclusions. In a study carried out in 80 public, private, and mixed centers by the Francisco de Vitoria University and the Complutense University of Madrid, it was established that half of the teachers had a very poor or a poor level regarding the use of technology and, at the same time, there were great differences between the digital competences of the mixed centers compared to the public centers, with the latter having worse results [74]. These findings should help the responsible institutions reflect on how they can make the same level of opportunities possible for every student in our increasingly digital world.

Therefore, the effective development of a quality education (SDG 4) that reduces inequalities (SDG 5, SDG 10) requires us to place people at the center of solutions (SDG 3, SDG 8) and avoid the danger of exploiting online learning only from economical perspectives. For this reason, both the diagnosis that we carried out and the possible solutions that the results have allowed us to outline respond to the model that we believe in, which places teachers and students at the center of the teaching-learning process. It is not only a matter of using technological tools but also of thinking digitally and respecting the technical, cognitive, and socio-emotional dimensions, as well as putting technology at the service of pedagogy as advocated by models such as TPACK [75] or its TPeCS review [76]. Any pedagogical alternative to improve the Digital Competence of Educators must be carried out through different practices where the training in the educational centers will go through different stages of both technical and conceptual appropriation of the technology [77,78]. The resilient response to any future situation that may arise should be composed of a suisection infrastructure and innovation (SDG 9) at the service of competent professionals. In short, next time we should be able to offer quality distance learning teaching instead of ERT, and the development of DCE is a fundamental axis for the sustainable and resilient education system we need.

**Author Contributions:** Conceptualization, J.P., U.G., and E.T.; data curation, J.P.; formal analysis, E.T. and N.B.; funding acquisition, U.G.; investigation, U.G.; methodology, U.G. and N.B.; project administration, U.G.; resources, E.T. and N.B.; software, J.P.; supervision, J.P. and U.G.; validation, J.P. and U.G.; writing—original draft, J.P. and U.G.; writing—review and editing, E.T. and N.B. All authors have read and agreed to the published version of the manuscript.

**Funding:** This research was funded by the University of the Basque Country, grant number GIU 19/010, PPGI19/11 and by the Basque Government, grant number IT1195-19.

**Conflicts of Interest:** The authors declare no conflict of interest. The funders had no role in the design of the study; in the collection, analyses, or interpretation of data; in the writing of the manuscript, or in the decision to publish the results.

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
