# Peer review of "Self-Perception of the Digital Competence of Educators during the COVID-19 Pandemic: A Cross-Analysis of Different Educational Stages"

_sustainability, doi:10.3390/su122310128_

Round 1

Reviewer 1 Report

My comments are in three parts: concept, inquiry, and trivia.

In general this study was well done. The Concept:  This is a teachers in digital environment perception study.  No problem!  But this journal focuses on sustainability education (which is acknowledged and developed in the intro). The link between SDG and public education is established.  The link between COVID and RET (the digital platform) is established.  However, the link between RET and SDG is NOT established either by argument, evidence, or rationale.  This is particularly egregious in the Discussion section.  It is MIA (missing in action!)  So, although I am not a journal editor, I am not sure this study is a "fit" for this journal in terms of content and despite its internal merit.

The Inquiry:  The population/sample:  Data are reported on the convenience sample.  However, the authors do not claim the sample is representative of the Basque Country.  And I have no idea of those population stats.

A variable named "emotions" is named and that word is used as a label throughout.  Rather, I think the variable is perceived emotion as the instrument items call for teachers' thinking about their emotions.

Ris an important stat.  But the reported R values seem relatively low, and nothing about the R values in similar studies is reported.

The readability of Tables 2 and 3 can improved with BOLD category labels and indenting sub-components.

Line 48 reports worldwide internet connectivity.  But data about Basque Country connectivity is not reported.  This is important.  I cannot guess this!

Line 236.  The authors claim that older teachers are less competent with educational technology.  Hummm!  I do not have the questions asked of the teachers.  But, maybe the older teachers compare their perceived tech expertise with their perceived face-to-face instruction.  Further, maybe older science teachers understand the complexity of inquiry and project instruction in a digital format.  The digital environment can and is reductive and limiting to the experienced, talented classroom instructor.  The authors' discussion here is NOT nuanced, and it may be over over simplified.  Houston, we have a problem!

The Trivia:

Line 33 makes a major claim; it need citation.

Line 106 Write out name of variable; then use acronym in parentheses.

Line 122. Write the number "five" (delete 5).  You did this in Line 129.

Line 133. The verb "compounded" and "size" are questionable.  Think there are better verbs.

Line 135.  The verb "was" is missing before "generated".

Line 155.  "Concerted centers"??  I have no idea.  Term is used elsewhere.

Line 199.  Sentence run-on.  Needs a period after ERT.

Line 203.  Needs a period after distribution.

Line 278.  Add Table number.

Line 339.  Verb "presented"???  Maybe a better verb here.

Author Response

Dear reviewer,

Thank you very much for your thorough revision. We have focused on the aspects you pointed out and we have made our best to improve the paper in that way. 

The Concept:  This is a teachers in digital environment perception study.  No problem!  But this journal focuses on sustainability education (which is acknowledged and developed in the intro). The link between SDG and public education is established.  The link between COVID and RET (the digital platform) is established.  However, the link between RET and SDG is NOT established either by argument, evidence, or rationale.  This is particularly egregious in the Discussion section. 

We have included the following new paragraphs in the Discussion section:

  • The pandemic has exposed that educational systems around the world should improve their resilience to unexpected situations for the effective development of "Quality Education" (SDG 4) even in those situations. Regarding the “Good health & well-being” (SDG 3) and “Decent work” (SDG 8) of educators, this research shows they have experienced high workload, stress and negative emotions during the ERT. This study also verifies that digital gaps have flourished in this emergency situation. Education systems should guarantee that the digital divide is reduced at all stages of education, which is linked to “Gender equality” (SDG 5) and “Reduced inequalities” (SDG 10).  
  • Therefore, the effective development of a quality education (SDG 4) that reduces inequalities (SDG 5, SDG 10) requires to place persons at the centre of solutions (SDG 3, SDG 8) and avoid the danger of exploiting online learning only from economical perspectives.
  • In short, next time we should be able to offer a quality distance learning teaching instead of ERT and the development of DCE is a fundamental axis for the sustainable and resilient education system we need. 

The Inquiry:  The population/sample:  Data are reported on the convenience sample.  However, the authors do not claim the sample is representative of the Basque Country. Line 48 reports worldwide internet connectivity.  But data about Basque Country connectivity is not reported.  This is important.  

New paragraph:

  • Finally, 4,589 responses to the questionnaire were obtained, that is, almost 10.5% of all the teaching staff of the Basque Country [49] participated voluntarily in the study. Regarding internet connectivity, 1.499.100 persons (80.2% of the population) in the Basque Country have a broadband connection [50].

A variable named "emotions" is named and that word is used as a label throughout.  Rather, I think the variable is perceived emotion as the instrument items call for teachers' thinking about their emotions.

We have introduced 'perceived' before emotions and workload labels in lines 168, 169, 186 and 187. We have not include it all along the text for simplicity and because we claim from the title that it is a 'self-perception' study. 

Ris an important stat.  But the reported R values seem relatively low, and nothing about the R values in similar studies is reported.

Three similar studies have been reported, as well as a reference for deeper explanation about the statistical analysis. Nevertheless, the whole description of results from hierarchical regressions has been rewritten. Some excerpts:

  • On the basis of meaningful relations among variables, a hierarchical regression can be designed in order to confirm the findings [58, 59, 60]. 
  • Hierarchical regression should be understood as a framework for model comparison [61]. In this sense, the prediction model that considers workloads and competencies as predictors is meaningfully better (ΔR2 = .15) than the first one, made of covariates.
  • The model explains 38% (total R2) of the variance shared by all variables together, which is higher than in the case of positive emotions (26%). The influence of competencies and workload predictors (ΔR2 = .25) is nearly the double than covariates (R2 = .13)

Line 236.  The authors claim that older teachers are less competent with educational technology.  Hummm!  I do not have the questions asked of the teachers.  But, maybe the older teachers compare their perceived tech expertise with their perceived face-to-face instruction.  Further, maybe older science teachers understand the complexity of inquiry and project instruction in a digital format.  The digital environment can and is reductive and limiting to the experienced, talented classroom instructor.  The authors' discussion here is NOT nuanced, and it may be over over simplified. 

New paragraph introduced in line 343, in age gap discussion:

  • Anyway, this article presents results based on the self-perception of teachers and this fact leaves room for interpretation. Maybe elder teachers feel their tech expertise is not enough in order to develop in a digital environment the complexity of inquiry and project instruction they can manage in a face-to-face environment thanks to their experience. This opens an interesting line for future research work.

Other improvements:

The readability of Tables 2 and 3 can improved with BOLD category labels and indenting sub-components. Done

Line 33 makes a major claim; it need citation. New citation and different wording...

UN claimed a decade of action:  “2020 needs to usher in a decade of ambitious action to deliver the Goals by 2030” [1]. Instead, the unexpected emergence of the COVID-19, and the resulting global confinement since March 2020 has contributed to hindering its development [2].

Line 106 Write out name of variable; then use acronym in parentheses. Done

Line 122. Write the number "five" (delete 5).  You did this in Line 129. Done

Line 133. The verb "compounded" and "size" are questionable.  Think there are better verbs. 'consisted of' and 'evaluate'

Line 135.  The verb "was" is missing before "generated". Done

Line 155.  "Concerted centers"??  I have no idea.  Term is used elsewhere. 'Private' used instead. The nuance of so-called 'concerted centers' in Spain is not relevant for the purpose of the paper.

Line 199.  Sentence run-on.  Needs a period after ERT. Done

Line 203.  Needs a period after distribution. Done

Line 278.  Add Table number. Done

Line 339.  Verb "presented"???  Maybe a better verb here. 'Had' instead

Thank you for your feedback. 

Best regards

Reviewer 2 Report

This research paper is very well presented, the study and method well described, literature detailed throughout - basically a paper ready for press, in my opinion. The topic of Covid-19 pandemic Emergency Remote Teaching is very important to study and publish findings in order to better prepare pre-service teachers, and develop policy to support inservice teachers for ERT type events. While the pandemic will eventually end, the need for highlighting described inequities in the ability of the education system to provide quality online and blended learning allows policy makers and even faculties of education a better understanding of what is needed for future digital age educators.

Excellent design, analysis, and well supported conclusions are presented in this paper - I certainly will be one academic who will be citing this research.

Well done - publish as soon as possible...

Dr Nathaniel Ostashewski

Author Response

Dear Dr Nathaniel Ostashewski

Thank you very much for your kind review!

Your warm words are very encouraging for us. We hope this pandemic comes to an early end and all the research can be somehow applied for achieving a more resilient Education System.

Kind regards

Reviewer 3 Report

Smartly relevant framing this around the SDGs and ERT (glad to see ERT, by the way, a term coined by Educause colleagues this past spring, clarifying that ERT is not "distance learning"). This enhances relevance. Although on p. 3 it states Remote Emergency Teaching. Please be consistent and use ERT throughout. I'd like to see more in the lit review about how ERT is not distance learning i.e., https://er.educause.edu/articles/2020/3/the-difference-between-emergency-remote-teaching-and-online-learning This is a minor fix.

Overall, this is an excellent paper. Important research and clear to read as well. The data collection and analysis are appropriate. 

Author Response

Dear reviewer

Thank you very much for your kind review!

We have fixed the typo you mentioned in page 3 ('Remote Emergency Teaching' instead of 'Emergency Remote Teaching'). We have looked for similar typos and we have fixed another one in page 2. 

Following your recommendation, we have added more references in the literature review about how ERT is not distance learning:

  • Portillo, S., Castellanos, L., Reynoso, O., & Gavotto, O. (2020). Enseñanza remota de emergencia ante la pandemia Covid-19 en Educación Media Superior y Educación Superior. Propósitos y Representaciones, 8 (SPE3), e589. Doi: http://dx.doi.org/10.20511/pyr2020.v8nSPE3.589
  • Galindo, Diana, García, Lorena, García, Rubén, González, Patricia, Hernández, Pablo C., López, Mireya, Luna, Verónica & Moreno, Carlos I. (2020). Recomendaciones didácticas para adaptarse a la enseñanza remota de emergencia. Revista Digital Universitaria (rdu), 21(5). doi: http://doi.org/10.22201/ cuaieed.16076079e.2020.21.5.15
  • Bozkurt, A., & Sharma, R. C. (2020). Emergency remote teaching in a time of global crisis due to CoronaVirus pandemic. Asian Journal of Distance Education, 15(1), i-vi. https://doi.org/10.5281/zenodo.3778083
  • Toquero, C. M. (2020). Emergency remote education experiment amid COVID-19 pandemic. IJERI: International Journal of Educational Research and Innovation, (15), 162-172. https://doi.org/10.46661/ijeri.5113

Kind regards!

Reviewer 4 Report

Very interesting and current paper that addresses a problem, the digital competence of educators, which, although always important, with the Covid-19 crisis has become the main topic of discussion today.
The results, with a very significant sample, show striking but worrying realities, pointing the direction to follow to reduce the deficiencies found in the teachers digital competence.
It has recent bibliography to support the study, although only one other is missing to support the statement made on lines 320 and 321, when it says "The data from this study corroborate what has been observed in other geographical areas, as well as other spheres outside of education".

Author Response

Dear reviewer

Thank you very much for your kind revision.

We tried to fix the missing reference you pointed out.

A reference is missing to support the statement made on lines 320 and 321, when it says "The data from this study corroborate what has been observed in other geographical areas, as well as other spheres outside of education".

Now it says:

The data from this study corroborate what has been observed in other geographical areas [64, 65], as well as other spheres outside of education [66].

  • López, J., Pozo, S., y Fuentes, A. (2019). Analysis of electronic leadership and digital competence of teachers of educational cooperatives in Andalucia (Spain). Multidisciplinary Journal of Educational Research, 9(2), 194-223. Doi: https://doi.org/10.4471/remie.2019.4149
  • Cabezas, M., Casillas, S., Sanches-Ferreira, M., y Teixeira, F.L. (2017). Do gender and age affectthe level of digital competence? A study with University students. Journal of Communication, (15), 115-132. doi: https://doi.org/10.14201/fjc201715115132en
  • Sáinz, M., Arroyo, L., y Castaño, C. (2020). Mujeres y digitalización. De las brechas a los algoritmos. Instituto de la Mujer y para la Igualdad de Oportunidades. Ministerio de Igualdad. Retrieved from https://cpage.mpr.gob.es accessed 05/11/2020